# Magnetostriction-polarization coupling in multiferroic $Mn_2MnWO_6$

Man-Rong Li [1,2], Emma E. McCabe [3], Peter W. Stephens[4], Mark Croft[5], Liam Collins[6], Sergei V. Kalinin [6], Zheng Deng[2], Maria Retuerto [2], Arnab Sen Gupta[7], Haricharan Padmanabhan[7], Venkatraman Gopalan[7], Christoph P. Grams [8], Joachim Hemberger[8], Fabio Orlandi [9], Pascal Manuel[9], Wen-Min Li[10], Chang-Qing Jin[10], David Walker[11] & Martha Greenblatt[2]

Double corundum-related polar magnets are promising materials for multiferroic and magnetoelectric applications in spintronics. However, their design and synthesis is a challenge, and magnetoelectric coupling has only been observed in $Ni_3TeO_6$ among the known double corundum compounds to date. Here we address the high-pressure synthesis of a new polar and antiferromagnetic corundum derivative $Mn_2MnWO_6$, which adopts the $Ni_3TeO_6$-type structure with low temperature first-order field-induced metamagnetic phase transitions ($T_N = 58\,K$) and high spontaneous polarization (~ 63.3 $\mu C\cdot cm^{-2}$). The magnetostriction-polarization coupling in $Mn_2MnWO_6$ is evidenced by second harmonic generation effect, and corroborated by magnetic-field-dependent pyroresponse behavior, which together with the magnetic-field-dependent polarization and dielectric measurements, qualitatively indicate magnetoelectric coupling. Piezoresponse force microscopy imaging and spectroscopy studies on $Mn_2MnWO_6$ show switchable polarization, which motivates further exploration on magnetoelectric effect in single crystal/thin film specimens.

[1] Key Laboratory of Bioinorganic and Synthetic Chemistry of Ministry of Education, School of Chemistry, Sun Yat-Sen University, Guangzhou 510275, P. R. China. [2] Department of Chemistry and Chemical Biology, Rutgers, the State University of New Jersey, 610 Taylor Road, Piscataway, NJ 08854, USA. [3] School of Physical Sciences, University of Kent, Canterbury, Kent CT2 7NH, UK. [4] Department of Physics & Astronomy, State University of New York, Stony Brook, NY 11794, USA. [5] Department of Physics and Astronomy, Rutgers, the State University of New Jersey, 136 Frelinghusen Road, Piscataway, NJ 08854, USA. [6] Centre for Nanophase Material Science & Institute for Functional Imaging of Materials, Oak Ridge National Laboratory, Oak Ridge, TN 37831, USA. [7] Department of Materials Science and Engineering, Pennsylvania State University, University Park, PA 16802, USA. [8] II Physikalisches Institut, Universität zu Köln, D 50937 Köln, Germany. [9] ISIS facility, STFC, Rutherford Appleton Laboratory, Chilton, Didcot, Oxfordshire OX11 0QX, UK. [10] Institute of Physics, Chinese Academy of Sciences, P. O. Box 603, Beijing 100080, China. [11] Lamont Doherty Earth Observatory, Columbia University, 61 Route 9W, PO Box 1000, Palisades, NY 10964, USA. Correspondence and requests for materials should be addressed to M.-R.L. (email: limanrong@mail.sysu.edu.cn) or to M.G. (email: greenbla@chem.rutgers.edu)

The structural features of corundum derivatives provide an ideal platform for designing polar and magnetic compounds, since magnetic ions can be incorporated into both the octahedral $A$- and $B$-sites to lead to strong magnetic interactions, accompanied by large spontaneous polarization ($P_S$) if the polar $LiNbO_3$ (LN, $R3c$), ordered ilmenite (OIL, $R3$), or $Ni_3TeO_6$ (NTO, $R3$) type structure is adopted[1–14]. Remarkable physical properties, such as multiferroic, piezoelectric, pyroelectric and second harmonic generation (SHG) effect, have been demonstrated in these materials. For example, the coexistence of weak ferromagnetism and ferroelectricity has been observed in the high-pressure LN-type $FeTiO_3$ and in recently-reported $GaFeO_3$[2, 14], and non-hysteretic colossal magnetoelectricity was found in collinear antiferromagnetic (AFM) NTO, which is, to the best of our knowledge, the only experimentally observed magnetoelectric coupling in the double corundum family[5]. In contrast to the off-centering displacement of $d^0$ $B$ cations in the octahedra occurring in many ferroelectric perovskites[15–17], in corundum-type $ABO_3$ or $A_2BB'O_6$ materials, the polarization reversal is driven by the small $A$ or $B$ cations moving between oxygen octahedra[18, 19], hence the $d^0$ configuration is not required. Therefore, considering the potential combinations of $A1$, $A2$, $B$, and $B'$ in the $(A1A2)BB'O_6$ corundum family (where $A1$, $A2$, and $B$, or any two of them could be the same element, or all cations could be different), a very large number of new multifunctional materials are anticipated with the assistance of high pressure (HP) synthesis techniques. However, to the best of our knowledge, only 14 polar and magnetic $A_2BB'O_6$-type corundum-related compounds have been reported to date (see Supplementary Table 1); of these, 11 compounds were experimentally prepared: $Ni_3TeO_6$[5, 20], $Ni_2ScSbO_6$[3], and $Ni_2InSbO_6$[3] were synthesized at ambient pressure, while the rest can only be stabilized at HP. $Zn_2FeOsO_6$[21] and $A_2FeMoO_6$ ($A$ = Sc, Lu)[22] have only been predicted by first principle calculations to show considerable large $P_S$.

In this work, we present the HP synthesis of a new polar and magnetic compound $Mn_2MnWO_6$ ($Mn^A_2Mn^BW^{B'}O_6$), which is predicted by first principles calculations to show switchable polarization in an anticipated ferrimagnetic ground state[23, 24]. The crystal and magnetic structures, cationic oxidation states as well as the physical properties, including second harmonic generation (SHG), magnetic properties, piezo-, pyro-, ferroelectric and magnetoelectric responses, are extensively studied.

## Results

**Crystal structure of $Mn_2MnWO_6$.** Earlier, $Mn_2MnWO_6$ single crystals prepared by $CO_2$-LASER technique in $H_2$-atmosphere at ambient pressure were reported with the $Mg_3TeO_6$-type structure (Supplementary Fig. 1)[25]. The $Mn_2MnWO_6$ polymorph we report here, prepared at 1673 K and 8 GPa (see Methods), forms in a different crystal structure. Synchrotron powder x-ray diffraction (SPXD) and neutron powder diffraction (NPD) data collected on the as-made sample indicate a rhombohedral ($R3$, No. 146) majority phase with a small impurity. The phases were identified as a NTO-type $Mn_2MnWO_6$ main phase ($a$ = 5.32323(3) Å, $c$ = 14.0589(1) Å, $V$ = 345.01(1) Å$^3$) and ~ 3.3(1) wt %-$MnWO_4$ wolframite[26, 27], from combined Rietveld refinements of SPXD and NPD data (Supplementary Fig. 2, $R_p/R_{wp}$ = 4.74/4.55%, $\chi^2$ = 3.67). All the cation sites were set as fully occupied, since free refinements lead to less than 1% deviation. There is good contrast between Mn and W neutron scattering lengths (Mn = −3.73 fm, W = 4.86 fm)[28] and allowing Mn − W antisite disorder (with constraints to maintain stoichiometry) in combined SPXD and NPD refinements suggested no disorder (antisite occupancies refined to < 1% with no improvement in fit). The final refinement

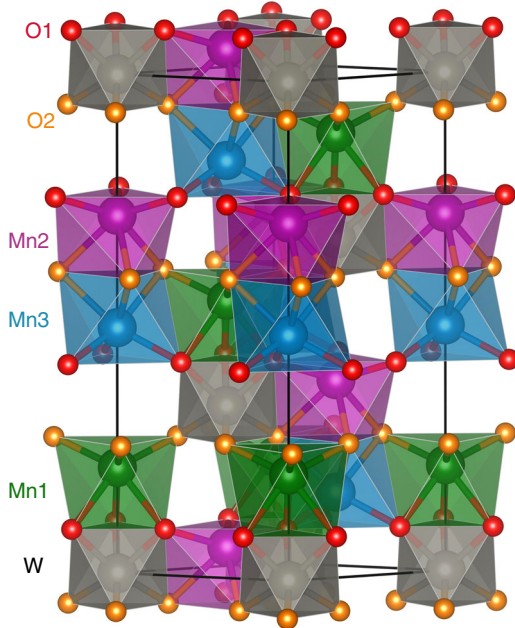

**Fig. 1** Crystal structure of $Mn_2MnWO_6$. The three-dimensional framework structure of $Mn_2MnWO_6$ viewed along [110] direction, to show the face-sharing $Mn1O_6$-$WO_6$ and $Mn2O_6$-$Mn3O_6$ octahedral pairs along the $c$-axis and the edge-sharing $Mn1O_6$-$Mn3O_6$ and $Mn2O_6$-$WO_6$ octahedral pairs in the $ab$-plane. The color codes of spheres are corresponding to Mn1-green, Mn2-purple, Mn3-cyan; W-light gray, O1-red, and O2-orange

results are listed in Supplementary Table 2 and the crystal structure is shown in Fig. 1. $Mn_2MnWO_6$ is isostructural with $Mn_2FeWO_6$ and crystallizes in NTO-structure with three independent Mn- (Mn1, Mn2, and Mn3), one W-, and two oxygen sites (O1 and O2), giving the structural formula of $(Mn1Mn2)^AMn3^BW^{B'}O_6$. The face-sharing $Mn1O_6$-$WO_6$ and $Mn2O_6$-$Mn3O_6$ octahedral pairs are arranged alternatively along the $c$-axis and separated by octahedral vacancies. In the $ab$-plane, the edge-sharing $Mn1O_6$-$Mn3O_6$ and $Mn2O_6$-$WO_6$ octahedral layers are connected alternatively to form a framework structure (see Fig. 1).

The paired face-sharing arrangement yields high octahedral distortions as reflected by the octahedral distortion parameter ($\Delta_M$)[29] and atomic displacement ($d_M$, distance between cation and its octahedral centroid) along the $c$-axis (Supplementary Table 3). The largest $\Delta$ and $d$ values are observed at the Mn2 site with $\Delta_{Mn2} = 5.07 \times 10^{-3}$, $d_{Mn2} = 0.517$ Å, which are very comparable to those of the Mn2 site ($5.45 \times 10^{-3}$ and 0.544 Å) in the $Mn_2FeWO_6$ analog[7]. These anisotropic atomic displacements induce a large $P_S$ (e.g. 63.3 $\mu C \cdot cm^{-2}$ at 290 K, as estimated by the point-charge displacement model)[30, 31], and give three long and three short metal-oxygen bond distances for each octahedron, varying from 2.061(1) to 2.377(2) Å for Mn-O and 1.867(2) to 1.999(1) Å for W-O. The average < Mn-O > distance lies between 2.196(2) and 2.217(2) Å, comparable to the < Mn-O > of 2.187(9) and 2.228(6) Å for Mn1 and Mn2 in $Mn_2FeWO_6$. The < W-O > value 1.933(9) Å is close to the < W-O > (1.925(9) Å) in $Mn_2FeWO_6$[7]. Bond valence sums (BVS) calculations[29, 32–34] give + 2.00, + 2.06, + 2.09, and + 5.84 for Mn1, Mn2, Mn3, and W, respectively, supporting formal cationic oxidation states of $Mn^{2+}_2Mn^{2+}W^{6+}O_6$ and well accounting for its slightly larger unit cell volume (345.01(1) Å$^3$, $r(^{VI}Mn^{2+})$ = 0.83 Å) than that of the isostructural $Mn^{2+}_2Fe^{2+}W^{6+}O_6$ (338.65(1) Å$^3$, $r(^{VI}Fe^{2+})$ = 0.78 Å)[35]. The large difference in ionic size and charge between $Mn^{2+}$ and $W^{6+}$ is

also responsible for the absence of anti-site disordering. The proposed formal cation oxidation states have been further evidenced by the x-ray absorption near edge spectroscopy (XANES) studies (Supplementary Figs. 3–5).

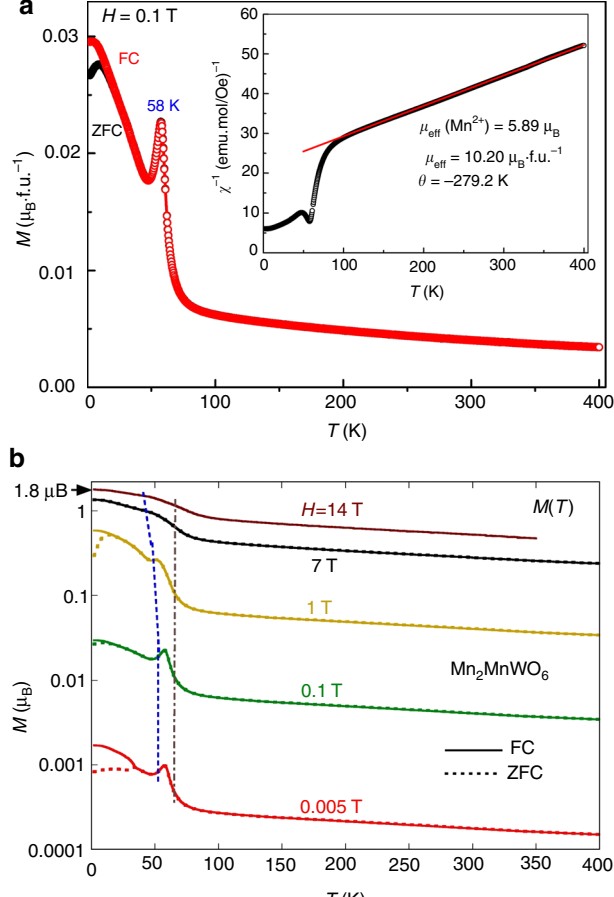

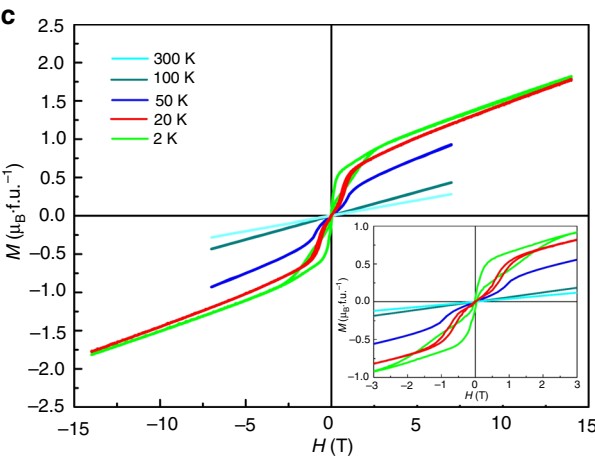

**Fig. 2** Magnetic properties of $Mn_2MnWO_6$. **a** Thermal evolution of the ZFC and FC mode magnetization ($M$) and the reciprocal susceptibility (inset) of $Mn_2MnWO_6$ measured with $H = 0.1$ T up to 400 K. **b** A logarithmic plot of the $M(T)$ curves at a series of magnetic fields between 0.005 and 14 T. The dashed line highlights the evolution of AFM transition temperatures. The dash-dot line highlights the presence of local magnetic correlations near 60 K in all finite magnetic fields. **c** Isothermal magnetization curves of $Mn_2MnWO_6$ measured at 2, 20, 50, 100, and 300 K between -14 and 14 T for 2 and 20 K; -7 and 7 T for 50, 100, and 300 K. Inset shows the curves between -3 and 3 T

**Magnetic properties of $Mn_2MnWO_6$.** The temperature-dependent magnetization $M(T)$ curves up to 400 K at 0.1 T (Fig. 2a) show that upon cooling the magnetization is enhanced below 80 K and a sharp AFM transition occurs at ∼ 58 K. Below 20 K, the zero-field cooling (ZFC) and field-cooling (FC) curves diverge, indicating a small ferromagnetic component or canted spins in an anisotropic system along with domain effects. At higher temperatures, $Mn_2MnWO_6$ follows the Curie − Weiss (CW) law; the negative Weiss temperature ($\theta = -$ 279.2 K) is much lower than the AFM transition at $T_N$ ∼ 58 K, again suggesting significant magnetic frustration/interaction. The effective magnetic moment ($\mu_{eff}$) derived from the CW fit of $1/\chi(T)$ over the paramagnetic region (inset of Fig. 2a) is 10.20 $\mu_B$·f.u.$^{-1}$ (f.u. = formula unit), which gives an average value of 5.89 $\mu_B$·f.u.$^{-1}$ for each Mn site, consistent with the theoretical value (5.92 $\mu_B$·f.u.$^{-1}$) of high-spin $d^5$-$Mn^{2+}$ state. Figure 2b shows the logarithmic-$M(T)$ curves collected in both ZFC and FC modes between 0.005 and 14 T up to 400 K. Below 1 T, the $M(T)$ plots manifest robust AFM transitions ∼ 58 K as evidenced by: the inflection point (below the peak) in the $M(T)$ data; the sharp peak in the d$M$/d$T$ curves shown in Supplementary Fig. 6; and the isothermal $M(H)$ hysteresis loops below 50 K in Fig. 2c and Supplementary Fig. 7. At 1 T, this AFM transition is weakened and moved to lower temperature as highlighted by the dashed line in Fig. 2b and Supplementary Figs. 6 and 8. Above 7 T this AFM order is substantially modified and the detailed character of the high field AFM state is uncertain. The $M(T)$ curves also evidence structure near 60 K at all fields indicating local magnetic correlations on this energy scale. The presence of a low temperature first-order field-induced metamagnetic phase transition (Supplementary Fig. 8), similar to that observed in $Mn_2FeWO_6$[7], is clear from the isothermal magnetization curves, $M(H)$, shown in Fig. 2c and in expanded views (with additional data) in Supplementary Fig. 7. The magnetization is far from saturation at 2 K and 14 T, and gives a value of only ∼ 1.82 $\mu_B$ f.u.$^{-1}$, indicating that AFM order still strongly constrains the field response in this regime.

**Magnetic structure of $Mn_2MnWO_6$.** To better understand the magnetic behavior of $Mn_2MnWO_6$, NPD data at lower temperatures were recorded. Additional Bragg reflections were observed below ∼ 55 K with intensity increasing smoothly on cooling (Supplementary Fig. 9). Some magnetic reflections were consistent with a magnetic unit cell commensurate with the nuclear crystal structure with magnetic propagation vector $k_1 = (0\ 0\ 3/2)$ (T point of the first Brillouin zone) while other reflections were broader and consistent with an incommensurate modulation with $k_2 \approx (0\ 0\ 0.3)$ (Λ line of the first Brillouin zone) (Supplementary Fig. 9). Good fits to the 5 K NPD data were obtained for models of $R_I\bar{3}(00\,g)t$ symmetry with the magnetic unit cell related to the nuclear cell through the transformation {(0-10)(110)(002)}. This magnetic superspace symmetry is a result of both the mT1 and mΛ2LE2 irreps acting on all three manganese sites. Models with only one irrep acting on each site gave poor fits to the data and unphysical moments for $Mn^{2+}$ sites. Various models with AFM ordering of manganese moments along [001] (described by irrep mT1) with these moments tilted towards the $ab$ plane rotating around [001] (described by mΛ2LE2 irrep), giving Mn moments arranged in cones around [001], gave good fits to the experimental data. Constraints were needed to give a stable refinement and convergence and the $z$ and $xy$ components of the Mn2 moments were constrained to be half and double those of the Mn1/Mn3 sites, respectively. This gives overall moments of 4.2(5) and 4.4(5) $\mu_B$ for Mn1/Mn3 and Mn2 sites at 5 K, respectively, with incommensurate propagation

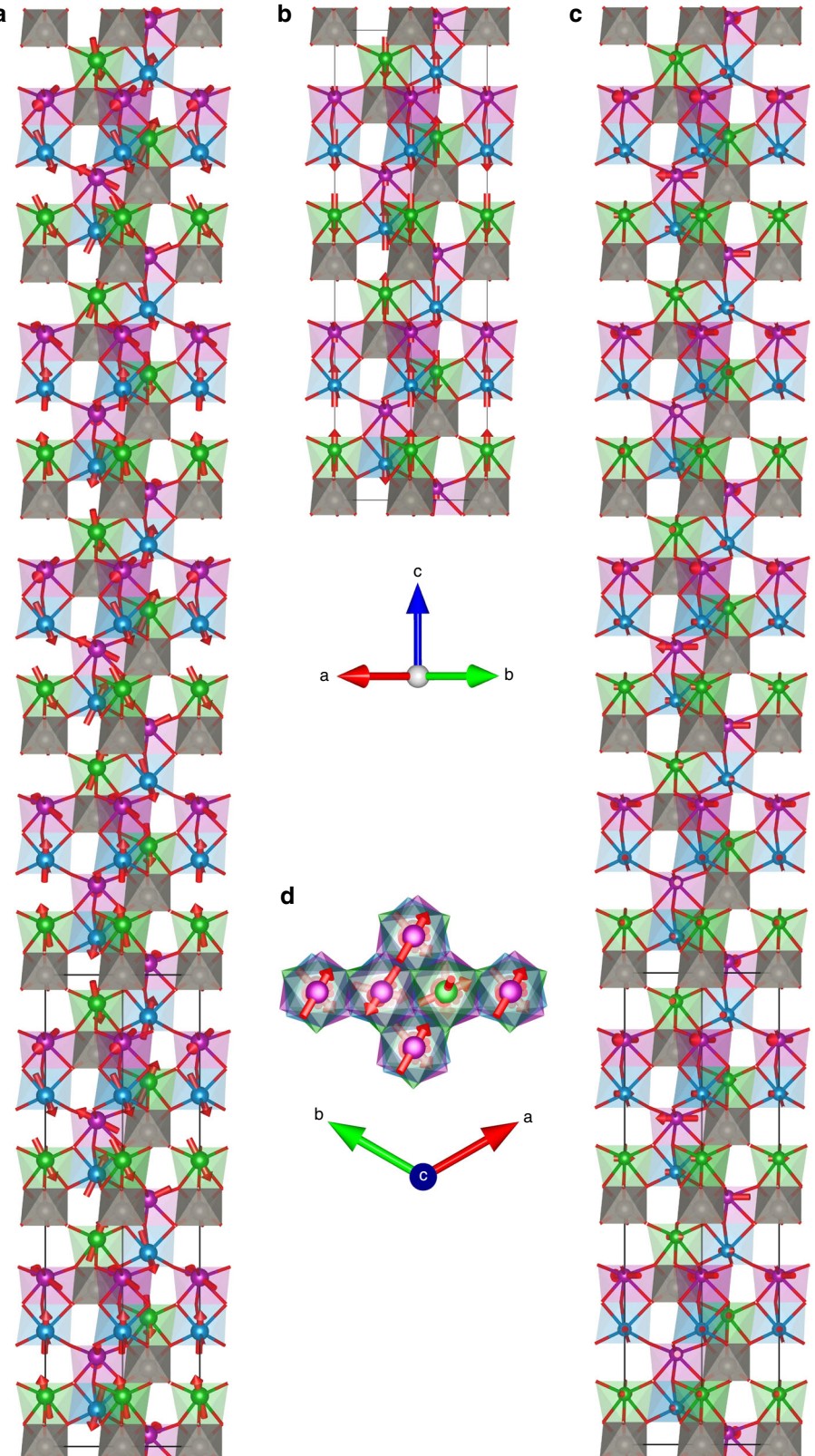

**Fig. 3** Illustration of the nuclear and magnetic structures of $Mn_2MnWO_6$ at 5 K. Mn1, Mn2, Mn3 and W sites and polyhedra are shown in green, purple, blue and grey, respectively, with Mn moments shown by red arrows (color online) (oxide ions are omitted for clarity). **a** shows the complete magnetic structure (showing six times the nuclear unit cell along *c*). **b** shows only the *z* component of Mn moments (described by commensurate mT1 irrep) and **c** shows only the *xy* component of Mn1 and Mn3 moments (described by mΛ2LE2 irrep) (showing six times the nuclear unit cell along *c*), also **d** showing view down along *c* axis of magnetic unit cell

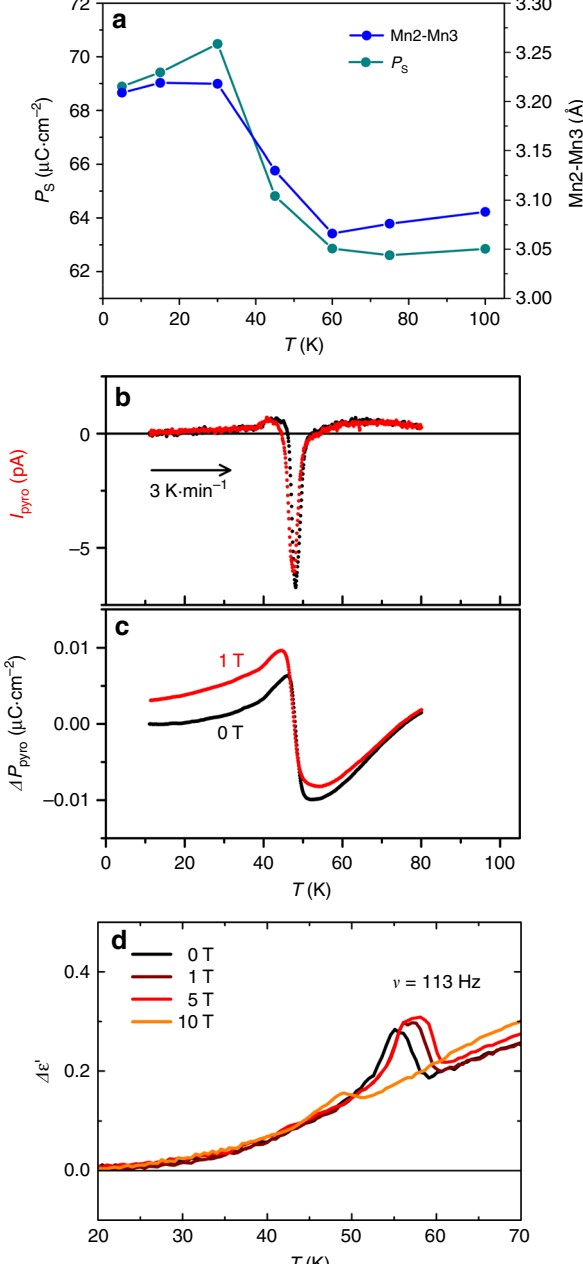

**Fig. 4** Magnetostriction-polarization coupling and pyroresponse in $Mn_2MnWO_6$. **a** Temperature dependent spontaneous polarization ($P_S$, calculated) and Mn2-Mn3 distance evolution in $Mn_2MnWO_6$ between 5 and 100 K. **b** Pyro-current as a function of temperature between 10 and 80 K, and **c** Pyroelectric polarization measured in 0 and 1 T upon warming and normalized to a common high temperature value. **d** Temperature dependent dielectric data between 0 and 10 T show anomalies around $T_N$ and indicate magnetoelectric coupling

vector $k = 0\ 0\ 0.6107(8)$ ($k = (0\ 0\ 0.305)$ with respect to the nuclear unit cell). We cannot rule out the possibility of similar magnetic structures giving equally good fits to the data, but the constraints imposed give almost equal moments for each manganese site and across the magnetic structure as might be expected for this insulating oxide. The magnetic and crystal structures at 5 K are shown in Fig. 3 and refinement profiles in Supplementary Fig. 10. Refinement details and selected bond lengths and distances are given in Supplementary Tables 4 and 5. Similar refinements were then carried out using short NPD scans

collected at selected temperatures on warming (with no magnetic component included above $T_N$).

This zero-field magnetic structure can be described as a superposition of commensurate AFM ordering along [001] (mT1 irrep) and a helical component in the $xy$ plane (mΛ2LE2 irrep) giving rise to the conical-AFM magnetic structure propagating along [001]. This magnetic structure can be thought of in terms of AFM coupling between Mn1 and Mn3 sites within the $Mn1Mn3O_3$ layers, with moments predominantly along [001]. The Mn2 moments are predominantly within the (001) planes in the opposite direction to the in-plane component in the $Mn1Mn3O_3$ layer directly above (this in-plane component is cancelled by that of other layers in the overall magnetic unit cell in this zero-field AFM structure). Manganese moments increase smoothly on cooling (Supplementary Figs. 11 and 12). The AFM arrangement of Mn1 and Mn3 moments predominantly along [001] within the $Mn1Mn3O_3$ layers satisfies the 90° super-exchange interactions expected to be AFM[36]. The Mn2 site is magnetically coupled to this $Mn1Mn3O_3$ layer via ~120° Mn1 − O2 − Mn2 interactions and ~86° Mn3 − O2 − Mn2 interactions across the shared face. Both these exchange interactions are likely to be AFM leading to magnetic frustration, consistent with magnetic susceptibility measurements described above. This frustration is somewhat relieved by the incommensurate modulation that reorients the moments away from [001], particularly for the Mn2 site, allowing its in-plane component to be oriented antiparallel to the in-plane component in the nearest $Mn1Mn3O_3$ layer (which lies directly above).

**Magnetostriction-polarization coupling and magnetoelectric effect in $Mn_2MnWO_6$.** The unit cell volume of $Mn_2MnWO_6$ decreases smoothly on cooling until the lowest temperatures when slight negative thermal expansion is observed (Supplementary Fig. 13). This is due to expansion of the unit cell along the [001] direction below $T_N$. This expansion is thought to be due to magnetostriction across the Mn2 − Mn3 face-shared poly-hedra: the Mn2 − Mn3 distance increases below the AFM ordering temperature as the Mn3 site moves towards the O1 layer and away from the O2 layer within the shared face (Fig. 4a and Supplementary Fig. 13), similar to structural changes observed in other materials containing $Mn_2O_9$ dimers[37]. This magnetostric-tion gives a dramatic increase of $P_S$ below $T_N$, in line with the increase of the Mn2 − Mn3 distance below $T_N$, giving computed $P_S$ of 62.86 μC·cm$^{-2}$ at 60 K and 70.48 μC·cm$^{-2}$ at 30 K (Fig. 4a and Supplementary Figs. 13 and 14). The coupling between spin structure and the lattice anomalies is well known to play an important role for the observation of multiferroicity[38]. The magnetostriction-polarization coupling around $T_N$ is also visible in the fluctuation of the SHG intensity (Supplementary Fig. 15). Figure 4b and c show the finite pyrocurrent and pyroelectric polarization response at 0 and 1 T, respectively. A clear anomaly/discontinuity can be detected in the pyrocurrent (Fig. 4b and Supplementary Fig. 16), the pyroelectric polarization (Fig. 4c) and the dielectric (Fig. 4d) curves in the vicinity of the magnetic transiton, qualitatively echoed by the magnetostriction effects, which couple the macroscopic polarization of the structure to magnetism. The small difference between pyroresponse at 0 and 1 T suggests possible magnetoelectric coupling, however, one should be aware of experimental uncertainties by lack of a robust effect (~ 0.05 μC·cm$^{-2}$ compared with the theoretical value of ~ 70 μC·cm$^{-2}$ at 20 K) in such a random-distribution polycrysalline specimen as also observed in the magnetic-field-dependent polarization measurement results in Supplementary Fig. 17. However, the temperature-dependent dielectric measurements at several magnetic fields from 0 to 10 T evidence anomalies around

$T_N$ in Fig. 4d. The shift of the transition temperature with the magnetic field as well as the observed suppression in high magnetic fields clearly convince magnetoelectric coupling in $Mn_2MnWO_6$.

**Switchable polarization of $Mn_2MnWO_6$.** To further explore the polar and ferroelectric properties of $Mn_2MnWO_6$, piezoresponse force microscopy (PFM) imaging and spectroscopic studies were performed at room temperature, since the surface deformation does not depend on the contact radius[39, 40], and hence is a direct measure of local piezoelectric properties[41, 42]. In conjunction with dual amplitude resonance-tracking (DART)[43], or band excitation (BE)[44, 45] modes, PFM allows to obtain quantitative information on material properties. The surface topography and PFM images of a polished sample embedded in epoxy is shown in Supplementary Fig. 18, with clearly visible variation of DART PFM contrast at the grain boundaries, some grains show clearly visible domain structures, highly reminiscent of domain structures for materials such as $BaTiO_3$[46, 47].

The switching properties of the material were explored with BE PFM polarization spectroscopy measurements[48]. The $750 \times 750$ nm region was first imaged by DART PFM as shown in Fig. 5a–c. Representative hysteresis loops of the amplitude and phase are shown in Fig. 5d. The clear hysteresis loops with the characteristic coercive biases of ~ 50 V are observed. Note that the loops are not saturated, suggesting that formation of domains are largely unstable and rapidly relax in the bias-off state. Here, the measurements are performed over rectangular grid of points ($35 \times 35$), giving rise to the 3D array of hysteresis loops. The latter can be processed to yield 2D maps of materials parameters such as coercive bias of polarization switching. The maps of remnant polarization for positive and negative coercive biases are shown in Fig. 5e and f, which bear some resemblance with underlying domain structure, suggesting the pinning of polarization by preexisting electroelastic fields. The final switching experiment was conducted on the region shown in Supplementary Fig. 19a. In this case, the surface is scanned by a strongly negatively (−100 V) biased tip within a 4 μm square, and subsequently with a strongly positively biased tip (+100 V) within a 2 μm square (Supplementary Fig. 19d). The polarization distributions after each poling measurement are shown in Supplementary Figs. 19b, c, e, f. Herein, it is conclusive that the polarization in $Mn_2MnWO_6$ is switchable, as further corroborated by the $P(E)$ loop measurements (Supplementary Fig. 20). For a quantitative image of the ferroelectric and magnetoelectric coupling effect, further exploration on single crystal sample is necessary.

**Comparison of $Mn_2MnWO_6$ and isostructural polar magnets.** It is relevant to compare the magnetic structure of $Mn_2MnWO_6$ with that of other magnetic NTO materials. In $Mn_2ScSbO_6$ and $Ni_2BSbO_6$ ($B$ = Sc, In), the non-magnetic ions create holes in the Mn/Ni magnetic sublattices preventing direct exchange between the magnetic sites[3, 6]. All these systems order AFM, but with no face-shared magnetic $M_2O_9$ ($M$ = magnetic cation) dimers or magnetic frustration, it is unlikely that magnetostriction-driven changes in polarization occur. However, it is interesting that without nearest-neighbor exchanges, $Ni_2BSbO_6$ ($B$ = Sc, In) is significantly more frustrated than $Ni_3TeO_6$ and adopts a non-collinear, helical magnetic structure with components of the $Ni^{2+}$ moments along both the $c$ direction and in the $ab$ plane[3]. NTO systems with three magnetic cations include $Mn_2FeWO_6$[7], $Mn_2FeMoO_6$[4], and $Ni_3TeO_6$[5, 49–52] and exhibit complex magnetic behavior. All three materials differ from $Mn_2MnWO_6$ (described here) in that they are reported to have collinear magnetic

structures with FM coupling between edge-shared magnetic sites within layers[49, 50]. The chiral, polar material $Ni_3TeO_6$[51] has been the most thoroughly characterized and it is useful to compare its behavior with that of $Mn_2MnWO_6$. Theoretical studies on $Ni_3TeO_6$ suggest that edge-linked Ni1 and Ni2 sites are coupled FM ($J_1$) and that face-linked Ni2 and Ni3 sites are also coupled FM ($J_2$). AFM $J_3$, $J_4$ and $J_5$ interactions couple the Ni3 site (analogous to the Mn2 site in $M_2MnWO_6$) to Ni1 and Ni2 sites in adjacent layers via corner-linked exchange; the relative strengths of these exchange interactions results in a small degree of frustration, and the experimentally observed (zero-field) magnetic structure is collinear with $Ni^{2+}$ moments oriented along [001][49, 50].

$Mn_2MnWO_6$ differs in that the Mn1 − Mn3 coupling between edge-linked sites is AFM. This leads to frustration in the coupling with the Mn2 site through face-shared coupling to Mn3 and corner-linked interactions with Mn1 and Mn3 sites, giving a higher degree of frustration in $Mn_2MnWO_6$ compared with $Ni_3TeO_6$ ($|\theta|/T_N \approx 5$ for $Mn_2MnWO_6$ and $\approx 1$ for $Ni_3TeO_6$[50]). This higher level of frustration is likely to give rise to the non-collinear magnetic structure of $Mn_2MnWO_6$ with a significant in-plane component for the Mn2 moment to somewhat relieve this frustration. Oh et al. reported interesting magnetic field dependent behavior for $Ni_3TeO_6$, with an increasing magnetic field along [001] able to switch the system from a higher polarization state to a state with lower polarization[5]. It is interesting that magnetostriction across the face-shared $M_2O_9$ dimers gives rise to a noticeable change in polarization in both these NTO materials and our variable-temperature NPD experiment allows us to study the magnetic and structural changes through the magnetic phase transition, clearly illustrating this effect (Fig. 4a, Supplementary Figs. 12 and 13). Both $Ni_3TeO_6$ and $Mn_2MnWO_6$ are polar as a result of the cation arrangement in this corundum-derived structure type, but the magnetic order modifies the existing electrical polarization[5, 49–52]. In $Mn_2MnWO_6$ the magnetic transition is driven by the one dimensional mT1 irreducible representation with order parameter μ and by the two dimensional mΛ2LE2 with order parameter $\eta_1$, $\eta_2$. Since the electrical polarization ($P$) is already present in the parent structure, it is possible to derive the coupling between the polarization and the magnetic order parameters as the product of $P$ and the magnetic free energy invariant. In this way, the linear quadratic coupling $P(\mu^2 + \eta_1^2 + \eta_2^2)$ is obtained. This coupling term is consistent with the magnetostriction observed experimentally in the neutron diffraction data and is at the basis of the change of the polarization at $T_N$.

In $Ni_3TeO_6$, the field-dependent behavior is ascribed to a spin-flop transition that reorients moments to within the $ab$ plane above a critical field along the polar $c$ axis, $H_c$[5]. Field-dependent neutron scattering experiments on the more frustrated $Mn_2MnWO_6$ (which already has some in-plane component for the moments) would be of interest to understand if a similar explanation might explain the field-dependence observed in magnetic susceptibility measurements (Figs. 2b, c, Supplementary Figs. 7 and 8). Oh et al. describe how applying an electric field along the polar $c$ axis of $Ni_3TeO_6$ increases the polarization but decreases the magnetization along $c$[5], presumably due to the increased Ni2 − Ni3 separation (which weakens the FM $J_2$ interaction) and changes the balance between competing $J_1$, $J_2$ and $J_4$ interactions; with in-plane $J_1$ interactions relatively weak, this may be sufficient to cause reorientation of Ni3 sites as well as Ni1 and Ni2 sites[52]. Single crystal experiments on $Mn_2MnWO_6$ would be valuable to investigate its (anisotropic) magnetic and dielectric behavior fully. Thus, our combined structural and magnetic study highlights the potential for NTO materials

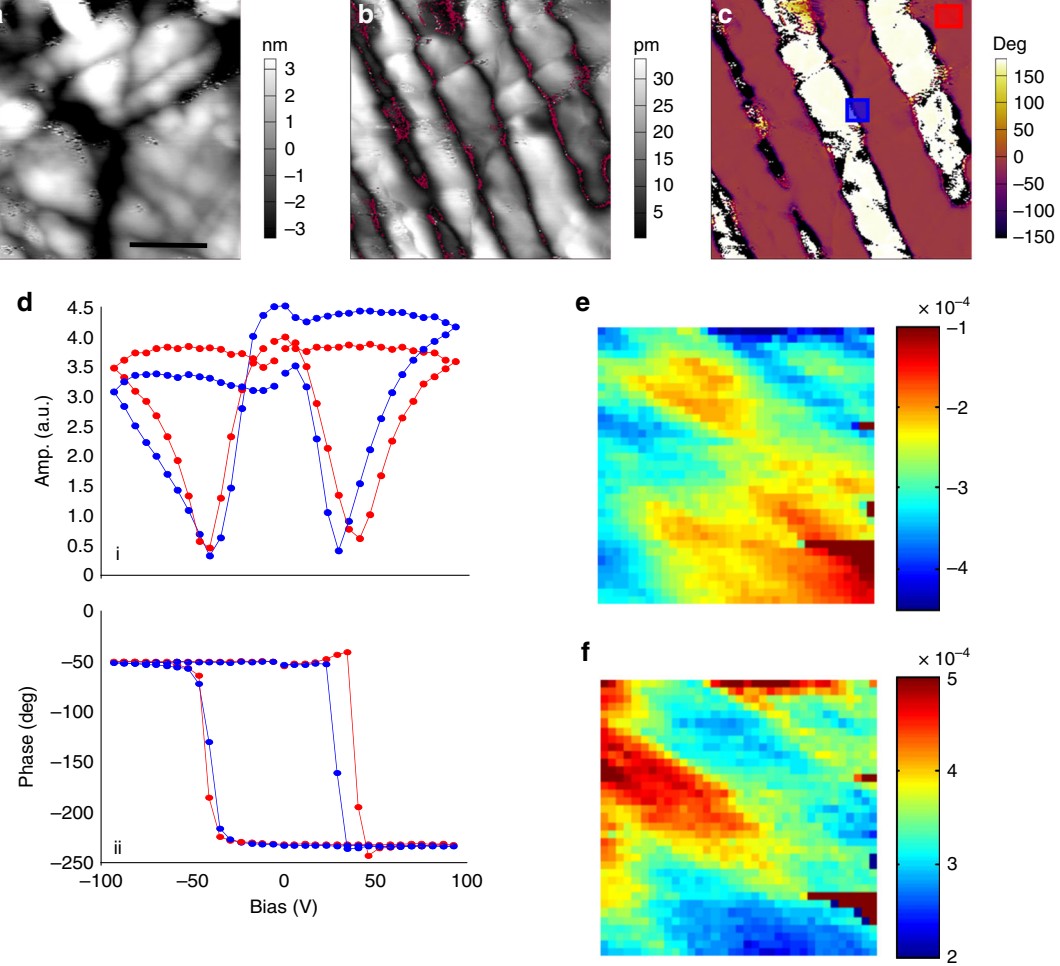

**Fig. 5** PFM results on $Mn_2MnWO_6$. **a** Topography (scale bar 250 nm). **b** PFM DART amplitude and **c** phase. **d** average amplitude (i) and phase (ii) BE PFM switching spectroscopy loops determined from square regions indicated in the phase image in (**c**). Remnant **e** negative and **f** positive amplitudes determined from fitting 35 × 35 grid measurement. Scale bar is the same for Fig. 5a–c

containing three magnetic cations to exhibit magnetostriction-influenced polarization changes and their complex field dependent behavior warrants further investigation to fully exploit their magnetoelectric coupling.

## Discussions

In summary, we have prepared, by high pressure-high temperature techniques, a corundum derivative phase $Mn_2MnWO_6$, which is a new polar and antiferromagnetic ($P_S$ ~ 63.3 $\mu C\cdot cm^{-2}$, $T_N = 58$ K) $Ni_3TeO_6$-type oxide with a low temperature first-order field-induced metamagnetic phase transition. The highly polarized spin structure shows antiferromagnetic coupling with magnetic moments predominantly along [001]. The magnetostriction-polarization coupling around the magnetic transition is echoed by the second harmonic generation effect and further corroborated by pyroresponse behavior with and without magnetic field, which, together with the magnetic-field-dependent polarization mesurements, qualitatively indicate magnetoelectric coupling. Piezoresponse force microscopy imaging and spectroscopy studies show that the polarization in $Mn_2MnWO_6$ is switchable, which motivates further exploration of ferroelectric and magnetoelectric coupling in single crystal and thin film specimens, as well as searching for new polar magnets in the corundum family.

## Methods

**Synthesis and crystal and magnetic structure determination**. Polycrystalline $Mn_2MnWO_6$ was prepared from a stoichiometric mixture of MnO (99.99%, Alfa Aesar) and $WO_3$ (99.8%, Alfa Aesar) at 1673 K under 8 GPa for 1 h in a Multi-Anvil Press as used in our previous work[7–9, 53]. SPXD data were recorded on beam line X-16C ($\lambda = 0.69991$ Å) at the Brookhaven National Synchrotron Light Source. Diffraction data analysis and Rietveld refinement were performed with the TOPAS software package[54, 55]. NPD data were collected on 0.1063 g sample (placed inside a 3 mm diameter vanadium can with sample height around 4 mm) at the ISIS Neutron source (Rutherford Appleton Laboratory (UK)) on the WISH diffractometer located at the second target station[56]. Data were collected at 290 K (~ 1 h) and then the sample was cooled to 5 K in an Oxford Instruments cryostat and a high quality data set was recorded (~ 3 h). Shorter scans were then collected on warming (~ 35 minute scans in 5 K increments to 80 K, then at 90 and 100 K). Rietveld refinements were carried out with Topas-Academic[54, 55] (for 290 K data) and Jana2006[57], for 5 K and intermediate temperature nuclear and magnetic structures). The magnetic symmetry analysis was carried out using ISO-DISTORT[58]. Double-frame data sets were collected at 5 and 100 K to confirm the presence of a magnetic Bragg reflection at ~ 47 Å below $T_N$. The Mn-K and W-$L_3$ XANES data were collected in both the transmission and fluorescence mode with simultaneous standards. All of the spectra were fitted to linear pre- and post-edge backgrounds and normalized to unity absorption edge step across the edge[4, 7, 8, 59, 60]. All of the XANES was performed on beam line X-19A at the Brookhaven National Synchrotron Light Source with a Si-111 double crystal monochromator.

**Magnetic properties measurements**. Magnetization measurements were carried out with a commercial Quantum Design superconducting quantum interference device (SQUID, up to 7 T) magnetometer and a physical property measurement system (PPMS, up to 14 T). The magnetic susceptibility was measured in zero-field-cooled (ZFC) and field-cooled (FC) conditions under 0.005-14 T magnetic

field, at temperatures ranging from $T = 5$–400 K. Isothermal magnetization curves were obtained at $T = 2$, 20, 50, 100 and 300 K under an applied magnetic field that varied from −14–14 T for 2 and 20 K and −7–7 T for 50, 100, and 300 K. Magnetization curves at additional temperatures and maximum fields are presented in Supplementary Figs. 6 and 7.

**SHG measurements**. The SHG experiments were performed in the reflection mode on polished pellets (cylinder pellet with 98(1) % of the theoretical density and ~ 2 mm of diameter and thickness of ~ 0.3 mm) of the as-made polycrystalline $Mn_2MnWO_6$. This is a widely used technique for determining noncentrosymmetry in materials[61–65]. This is an optical technique in which two photons with fields $E_j$ and $E_k$ of frequency $\omega$ and directions $j$ and $k$, respectively, interact with a material with a non-zero $d_{ijk}$ tensor (non-centrosymmetric) forming a polarization $P_i^{2\omega}$ (nonlinear) of frequency $2\omega$ in the $i$ direction. The SHG intensity, $I^{2\omega}$ is detected using a Hamamatsu photo multiplier tube. A Ti-sapphire laser (Spectra-Physics) with an output of 800 nm, 80 fs pulses at 2 kHz frequency was used for this experiment. Temperature scans were performed with an Oxford cryostat (50–320 K) and a user customized heater (298–800 K).

**Electric measurements**. The pyro-current was measured with an electrometer (Keithley 6517) at cooling/heating rates of 1–3 K·min⁻¹ in a PPMS Cryo-Magnet (Quantum Design); the corresponding polarization data was gained by numerical integration. The ferroelectric P-E loops and magnetic-field dependent polarization measurements were recorded with a modified Sawyer-Tower circuit employing a Keithley 6517 electrometer with linear field ramping at rates of 100 (V·mm⁻¹)·s⁻¹ and 100 Oe·s⁻¹. The samples were sandwiched between Ag paste-deposited electrodes, in parallel plate geometry for the above measurements. It should be noted that due to the use of polycrystalline samples a pyroelectric can only be expected for a not perfectly random distribution of structural domain orientations. Thus the polarization values gained only reflect the qualitative field and temperature dependence and have to be much smaller than the values gained from single crystals or structural refinement. In addition, thermal gradients on the sample may lead via piezo-coupling to finite charge contributions (tertiary pyro-effect) which constitute further uncertainties in the evaluated polarization. The dielectric properties were measured with a NovoControl-Alpha frequency response analyzer.

**PFM measurements**. In PFM, application of the periodic electric bias to the conductive scanning probe microscopy tip in contact with the surface results in the surface deformation, due to converse piezoelectric effect. This deformation is detected as the periodic deflection of the tip via microscope electronics. This approach has been broadly used for imaging ferroelectric domains in a broad range of ferroelectric and piezoelectric crystals, ceramics, and thin films[66–70]. The PFM measurements were performed at room temperature with 6 $V_{pp}$ ac bias applied to a Pt/Cr-coated probe (Budget sensors Multi75E-G). For PFM imaging the drive frequency of the ac bias was centred at the contact resonance (~ 350 kHz) and dual amplitude resonance tracking was then used to track the contact resonance as the tip was scanned across the sample surface[43]. For the polarization switching experiments a band of frequencies (~ 80 KHz) centered around the contact resonance frequency were excited, as an additional DC bias was swept from −90 to + 90 V. Extraction of the tip parameters were determined from fitting of the response to a simple harmonic oscillator model as described elsewhere[44, 45].

**Data availability**. The data that support the findings of this study are available from the corresponding authors on request.

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

## Acknowledgements

This work was supported by the NSF-DMR-1507252 grant. M.R.L. thanks the "One Thousand Youth Talents" Program of China. Use of the NSLS, Brookhaven National Laboratory was supported by the DOE BES (DE-AC02-98CH10886). A.S.G., H.P. and V.G. acknowledge support for them the Penn State NSF-MRSEC Center for Nanoscale Science grant (DMR-1420620). M.R. thanks the Spanish *Juan de la Cierva* grant FPDI-2013-17582. C.P.G. and J.H. were funded through the Institutional Strategy of the University of Cologne and CRC1238 within the German Excellence Initiative. Experiments at the ISIS Pulsed Neutron and Muon Source were supported by a beam-time allocation from the Science and Technology Facilities Council. P.M. and F.O. acknowledge support from the project TUMOCS. This project has received funding from the European Union's Horizon 2020 research and innovation programme under the Marie Sklodowska-Curie grant agreements No.645660. PFM experiments were conducted and partially supported (L.C., S.V.K.) at the Center for Nanophase Materials Sciences, which is a US DOE Office of Science User Facility. We are grateful to Prof J.S.O. Evans and Dr A. McLennan for making a long cif Topas macro widely available.

## Author contributions

M.-R.L. and M.G. conceived the idea of the project. M.-R.L., M.R., and D.W. prepared the samples. P.W.S. performed the SPXD work. M.C. did the XANES and magnetism analysis. E.E.M., F.O. and P.M. collected the NPD data and and analyzed the nuclear magnetic structures. M.-R.L. calculated the magnetostriction-polarization coupling effect. A.S.G., H.P., and V.G. conducted the SHG measurements. Z.D., W.-M.Li, and C.-Q.J. meausred the magnetic properites. L.C. and S.V.K. measured and analyzed the PFMpart. C.P.G. and J.H. measured the pyroelectric effect and magnetic-field-dependent polarization evolution. M.-R.L. and M.G. co-wrote the paper. All authors commented on the manuscript. M.G. supervised the project.

## Additional information

**Competing interests:** The authors declare no competing financial interests.

