## [Peer Review File · Nature Communications]

Reviewers' comments:

Reviewer #1 (Remarks to the Author):

Li et al have synthesized the corundum-type Mn_2MnWO_6 through high-pressure method and investigated its multiferroic properties in this manuscript. The central theme of this paper is to study the relationship between magnetostriction and polarization in this compound. Although the authors have given detailed characterizations in this manuscript, the physical mechanism for this relationship is not explained/discussed clearly and some experimental results still puzzle the reviewer.

1. Why the divergence between the ZFC and FC curves at 0.1 T is smaller than that of curves measured under 0.005 and 1 T?
2. The banana-shaped loops in Fig. S18 indicate that the sample isn't insulating enough and the ferroelectricity isn't intrinsic.
3. Why does the ferroelectric polarization decrease from 30 K?
4. Why the sign of pyroelectric current is negative?
5. The temperature dependence of pyrocurrent and pyroelectric polarization should be measured in many more magnetic fields that larger than 1 T.
6. The pyrocurrent peak emerges at magnetic transition temperature suggests that the partial ferroelectric polarization originates from spin configuration. The authors should give more detailed explanation on how the spin ordering induces the ferroelectric from the viewpoint of magnetic symmetry.

Reviewer #2 (Remarks to the Author):

The authors report their study of the structural, magnetic and dielectric properties of a new material synthesized under pressure. The compound adopts a corundum-related structure and shows interesting, possibly coupled, magnetic and dielectric properties.

The study is interesting as it adds a new compound to the family of "multiferroic" Ni_3TeO_6 -type systems. The report is convincing as 1) the authors have performed a wide variety of experiments to investigate in detail the nature and properties of the material, and 2) the analysis of the obtained data is sound. As such the manuscript is a very nice piece of work, and certainly deserves publication.

My only negative comment would be that in spite of this thoroughness, it is not clear why this material is particularly interesting, e.g. in comparison of Mn_2FeWO_6 studied in Ref. 7 by the authors using the exact same modus operandi. To highlight the novelty of the present findings and their relevance to designing new corundum-related multiferroics and/or understanding the structure-property relationship in this type of materials, I would suggest the authors to describe the similarities or differences of the magnetic and dielectric properties with those of the isostructural or related compounds listed in table S1, in the light of the magnitude of their respective crystallographic parameters (bond lengths, angles, etc).

A minor issue, how have the authors identified the presence of ferromagnetic correlations around 60 K? solely from the inflection in dM/dT (Fig. S6) or does it appear in the neutron data? in Ref. 7, such inflection was attributed to antiferromagnetic correlations.

Reviewer #1 (Remarks to the Author):

Li *et al* have synthesized the corundum-type Mn_2MnWO_6 through high-pressure method and investigated its multiferroic properties in this manuscript. The central theme of this paper is to study the relationship between magnetostriction and polarization in this compound. Although the authors have given detailed characterizations in this manuscript, the physical mechanism for this relationship is not explained/discussed clearly and some experimental results still puzzle the reviewer.

1. Why the divergence between the ZFC and FC curves at 0.1 T is smaller than that of curves measured under 0.005 and 1 T?

Authors' response

The relevant $M(T)$ curves are plotted on a linear scale in the two figures below. The apparent discrepancy is an artifact of the logarithmic magnetization scale. The two lower fields presumably reflect domain effects within the low-field *c*-AFM phase. The much larger effect at 1 T presumably reflects the difference between: FC preparation in the strongly canted *c*-AFM' phase; whereas the ZFC is at least partially prepared in the *c*-AFM phase which then transforms upon heating to the *c*-AFM' phase.

2. The banana-shaped loops in Fig. S18 indicate that the sample is not insulating enough and the ferroelectricity is not intrinsic.

Authors' response

We agree with the referee and apologize that we did not make this point more clearly in the manuscript. The double up-cycle (dotted line) demonstrates that at 200 K the sample is too conducting to resolve switchable polarization in the bulk. To clear this point up we amended the caption of Figure S18 (**now Figure S19 in the revised version**) in Page S28 of the Supplementary Information:

“Fig. S19 $P(E)$ loop measured on polycrystalline Mn_2MnWO_6 pellet between 10 and 150 K, showing very small switchable polarization (in the range of $0.005 \mu C/cm^2$). At 200 K the bulk conductivity dominates the polarization response of the sample: The dashed line is a double up-cycle of the electric field at the same temperature. The second up-cycle yields again a similar enhancement of polarization compared to the first which cannot be explained by the switching of intrinsic polarization but stems from ohmic contributions. However, at lower temperatures the residual conductivity of the sample decreases drastically and therefore this ohmic contribution vanishes leaving only the denoted remnant polarization.”

3. Why does the ferroelectric polarization decrease from 30 K?

Authors' response

The overall temperature dependence of the polarization stems from the pyroelectric background caused by the polar lattice structure. At the magnetic transition a small increase of the absolute value can be observed due to the onset of magnetic order and a therefore obvious magneto-electric coupling. At low temperatures, for which the magnetic order parameter is saturated, the overall T -dependence again is given by the pyroelectric background. Meanwhile, this variation can be quantitatively reflected by point-charge model calculations of the individually atomic P_S contribution at different temperatures, which show that the $\sim 1.59 \mu C/cm^2$ drop of the ferroelectric polarization below 30 K is mainly due to the faded contribution from O1/O2 and enhanced donation from the negative counterpart Mn3. Thus the following Figure S14 is added to Page S23 of the Supplementary Information:

“**Figure S14.** Temperature dependent individual atomic P_S contribution in Mn_2MnWO_6 between 5 and 100 K as calculated from point-charge-model method. The total net $P_S = P_S^+(\text{Mn1} + \text{Mn2} + \text{O1} + \text{O2}) - P_S^-(\text{Mn3} + \text{W})$ regarding the formal oxidation state and atomic displacement direction along the c -axis in **Fig. 1** ($P_S^+(\text{Mn1}) = P_S^+(\text{Mn2})$, $P_S^+(\text{O1}) = P_S^+(\text{O2})$ from point-charge-model calculations). The ‘+’ and ‘-’ represent the polarization directions denoted as blue (+) and red (-) symbols, respectively. ΔP_S^I and ΔP_S^{II} are for the P_S evolution between 5 and 30 K and 60 and 100 K upon cooling, respectively. P_S evolution of (a) Mn1/Mn2, O1/O2, Mn3, and W, and (b) Mn1/Mn2, (c) O1/O2, (d) Mn3, and (e) W. One should note that the polarization directions of (Mn1, Mn2, O1, and O2) and (Mn3 and W) are opposite. Apparently, the decreasing of $\Delta P_S^I(\text{O1/O2})$ ($1.158 \mu\text{C}/\text{cm}^2$, Figure S14c) and increasing of $\Delta P_S^I(\text{Mn3})$ ($0.579 \mu\text{C}/\text{cm}^2$ in Figure S14d) in opposite direction are mainly responsible for the total P_S drop of $1.59 \mu\text{C}/\text{cm}^2$ when cooling from 30 K. The P_S values are almost identical between 60 and 100 K ($0.01 \mu\text{C}/\text{cm}^2$ difference).”

4. Why the sign of pyroelectric current is negative?

Authors' response

As indicated by the arrow in panel b, of Figure 4, the measurement shown was taken during heating the sample. Therefore, the decay of the polarization with increasing temperature leads to a negative pyro-current as expected.

5. The temperature dependence of pyrocurrent and pyroelectric polarization should be measured in many more magnetic fields that larger than 1 T.

Authors' response

In principle the referee is right that in order to study the details of a magneto-electric coupling more measurements would be needed. However, as discussed in the paper the error bars are too large to give more than an estimation of the order of magnitude for the ME effect near the metamagnetic transition. Above 1 T the magnetic order is not altered anymore and the $M(H)$ data start to saturate. Therefore we would not expect further qualitative changes in the induced polarization. Furthermore, we want to remind that the absolute values for the polarization analysis are severely hampered by the polycrystalline nature of our samples and future investigations using single crystals or films are planned.

6. The pyrocurrent peak emerges at magnetic transition temperature suggests that the partial ferroelectric polarization originates from spin configuration. The authors should give more detailed explanation on how the spin ordering induces the ferroelectric from the viewpoint of magnetic symmetry.

Authors' response

We thank the reviewer for highlighting this lack of clarity in the manuscript. The inversion symmetry is broken by the cation ordering in this Ni_3TeO_6 -related crystal structure, separate from any magnetic behavior. However, it is interesting that the polarization along [001] is influenced by the magnetic order due to the magnetostriction in the face-shared M_2O_9 dimers, as observed for Ni_3TeO_6 . We've written a clearer description of this magnetic structure and of the possible coupling with the electrical polarization, in comparison to that reported for related materials, to clarify this and put this material in context (see response to reviewer 2 below).

Reviewer #2 (Remarks to the Author):

The authors report their study of the structural, magnetic and dielectric properties of a new material synthesized under pressure. The compound adopts a corundum-related structure and shows interesting, possibly coupled, magnetic and dielectric properties.

The study is interesting as it adds a new compound to the family of "multiferroic" Ni_3TeO_6 -type systems. The report is convincing as 1) the authors have performed a wide variety of experiments to investigate in detail the nature and properties of the material, and 2) the analysis of the obtained data is sound. As such the manuscript is a very nice piece of work, and certainly deserves publication.

My only negative comment would be that in spite of this thoroughness, it is not clear why this material is particularly interesting, e.g. in comparison of Mn_2FeWO_6 studied in Ref. 7 by the authors using the exact same modus operandi. To highlight the novelty of the present findings and their relevance to designing new corundum-related multiferroics and/or understanding the structure-property relationship in this type of materials, I would suggest the authors to describe the similarities or differences of the magnetic and dielectric properties with those of the isostructural or related compounds listed in table S1, in the light of the magnitude of their respective crystallographic parameters (bond lengths, angles, etc).

Authors' response

We thank the reviewer for these suggestions. Chemically and crystallographically, Mn_2MnWO_6 is isostructural with other NTO-type compounds in Table S1 like Mn_2FeWO_6 as compared in the last paragraph of *Crystal structure of Mn_2MnWO_6* section in *Results and Discussions* part in Page 4. We had neglected to properly put the comparison of physical properties in the context of related materials which may have lessened the apparent impact of the work. We discussed the magnetic and dielectric behavior of Mn_2MnWO_6 and fully compared it to related materials (including closely-related polar and chiral Ni_3TeO_6 , *Nat. Commun.* 2014 (cited 19

times). This improves the manuscript and clarifies why our study of this new material is an important step forward in this field. The following Section has been added to Line 7 of Page 10 in the revised manuscript:

“Comparison of Mn_2MnWO_6 and isostructural polar magnets. It is relevant to compare the magnetic structure of Mn_2MnWO_6 with that of other magnetic NTO materials. In $\text{Mn}_2\text{ScSbO}_6$ and Ni_2BSbO_6 ($B = \text{Sc, In}$), the non-magnetic ions create holes in the Mn/Ni magnetic sublattices preventing direct exchange between the magnetic sites.^{3,6} All these systems order AFM, but with no face-shared magnetic M_2O_9 ($M = \text{magnetic cation}$) dimers or magnetic frustration, it is unlikely that magnetostriction-driven changes in polarization occur. However, it is interesting that without nearest-neighbor exchanges, Ni_2BSbO_6 ($B = \text{Sc, In}$) is significantly more frustrated than Ni_3TeO_6 and adopts a non-collinear, helical magnetic structure with components of the Ni^{2+} moments along both the c direction and in the ab plane. NTO systems with three magnetic cations include Mn_2FeWO_6 ,⁷ $\text{Mn}_2\text{FeMoO}_6$,⁴ and Ni_3TeO_6 ^{5,49-52} and exhibit complex magnetic behavior. All three materials differ from Mn_2MnWO_6 (described here) in that they are reported to have collinear magnetic structures with FM coupling between edge-shared magnetic sites within layers.^{49,50} The chiral, polar material Ni_3TeO_6 ⁵¹ has been the most thoroughly characterized and it is useful to compare its behavior with that of Mn_2MnWO_6 . Theoretical studies on Ni_3TeO_6 suggest that edge-linked Ni1 and Ni2 sites are coupled FM (J_1) and that face-linked Ni2 and Ni3 sites are also coupled FM (J_2). AFM J_3 , J_4 and J_5 interactions couple the Ni3 site (analogous to the Mn2 site in Mn_2MnWO_6) to Ni1 and Ni2 sites in adjacent layers via corner-linked exchange; the relative strengths of these exchange interactions results in a small degree of frustration, and the experimentally observed (zero-field) magnetic structure is collinear with Ni^{2+} moments oriented along [001].^{49,50}

Mn_2MnWO_6 differs in that the Mn1 – Mn3 coupling between edge-linked sites is AFM. This leads to frustration in the coupling with the Mn2 site through face-shared coupling to Mn3 and corner-linked interactions with Mn1 and Mn3 sites, giving a higher degree of frustration in Mn_2MnWO_6 compared with Ni_3TeO_6 ($|\theta|/T_N \approx 5$ for Mn_2MnWO_6 and ≈ 1 for Ni_3TeO_6 ⁵⁰). This higher level of frustration is likely to give rise to the non-collinear magnetic structure of Mn_2MnWO_6 with a significant in-plane component for the Mn2 moment to somewhat relieve this frustration. Oh *et al.* reported interesting magnetic field dependent behavior for Ni_3TeO_6 , with an increasing magnetic field along [001] able to switch the system from a higher polarization state to a state with lower polarization.⁵ It is interesting that magnetostriction across the face-shared M_2O_9 dimers gives rise to a noticeable change in polarization in both these NTO materials and our variable-temperature NPD experiment allows us to study the magnetic and structural changes through the magnetic phase transition, clearly illustrating this effect (Figures 4a, S12 and S13). Both Ni_3TeO_6 and Mn_2MnWO_6 are polar as a result of the cation arrangement in this corundum-derived structure type, but the magnetic order modifies the existing electrical polarization.^{5,49-52} In Mn_2MnWO_6 the magnetic transition is driven by the

one dimensional mT1 irreducible representation with order parameter μ and by the two dimensional mA2LE2 with order parameter η_1, η_2 . Since the electrical polarization (P) is already present in the parent structure, it is possible to derive the coupling between the polarization and the magnetic order parameters as the product of p and the magnetic free energy invariant. In this way, the linear quadratic coupling $P(\mu^2 + \eta_1^2 + \eta_2^2)$ is obtained. This coupling term is consistent with the magnetostriction observed experimentally in the neutron diffraction data and is at the basis of the change of the polarization at T_N .

In Ni_3TeO_6 , the field-dependent behavior is ascribed to a spin-flop transition that reorients moments to within the ab plane above a critical field along the polar c axis, H_c .⁵ Field-dependent neutron scattering experiments on the more frustrated Mn_2MnWO_6 (which already has some in-plane component for the moments) would be of interest to understand if a similar explanation might explain the field-dependence observed in magnetic susceptibility measurements (Figures 2b, 2c, S7 and S8). Oh *et al.* describe how applying an electric field along the polar c axis of Ni_3TeO_6 increases the polarization but decreases the magnetization along c ,⁵ presumably due to the increased Ni2 – Ni3 separation (which weakens the FM J_2 interaction) and changes the balance between competing J_1, J_2 and J_4 interactions; with in-plane J_1 interactions relatively weak, this may be sufficient to cause reorientation of Ni3 sites as well as Ni1 and Ni2 sites.⁵² Single crystal experiments on Mn_2MnWO_6 would be valuable to investigate its (anisotropic) magnetic and dielectric behavior fully. Thus, our combined structural and magnetic study highlights the potential for NTO materials containing three magnetic cations to exhibit magnetostriction-influenced polarization changes and their complex field dependent behavior warrants further investigation to fully exploit their magnetoelectric coupling.”

A minor issue, how have the authors identified the presence of ferromagnetic correlations around 60 K? solely from the inflection in dM/dT (Fig. S6) or does it appear in the neutron data? in Ref. 7, such inflection was attributed to antiferromagnetic correlations.

Authors' response

Reviewer 2 makes a good and useful point here. There is evidence in the $M(H)$ and dM/dT results for an enhanced magnetic response near ~ 60 K at all fields. However, labeling this enhancement a FM contribution, we agree, is an overreach. In Ref. 7, in the ordered state, there was a field induced ferrimagnetic component introduced into the AFM state. The presence of an enhanced $M(T)$ field response at high temperatures could therefore reasonably be attributed to short range AFM correlations which had an uncompensated moment. In Mn_2MnWO_6 , while there is

spin canting, all moments are fully compensated in AFM state determined by the neutron scattering. For Mn_2MnWO_6 , the presence of a field induced transition to a higher magnetic response state at low temperature and the high temperature structure in the dM/dT curves are qualitatively similar to that observed in Ref. 7, however the $H = 0$ neutron diffraction results do not provide grounds to label the dM/dT structure to be due to FM correlations.. Accordingly, throughout the text we have removed references to the FM label. We have replaced it with simply “local magnetic correlations”. Finite field neutron diffraction will be required to properly address this issue.

In Line 2 of Page 6, we have changed “Above 7 T this AFM order is substantially modified with enhanced FM correlations and the vestigial AFM character uncertain.” to “Above 7 T this AFM order is substantially modified and the detailed character of the high field AFM state is uncertain. The $M(T)$ curves also evidence structure near 60 K at all fields indicating local magnetic correlations on this energy scale.”

In Line 9 of Page 6, “indicating that AFM order still strongly constrains the field response in this regime.” has been added.

In Page 21, “The dash-dot line highlights the presence of local magnetic correlations near 60 K in all finite magnetic fields.” has been added to Figure 2b caption.

Reviewers' comments:

Reviewer #1 (Remarks to the Author):

Even though the revised manuscript has corrected most of the comments, the evidences for magnetic induced polarization and magnetoelectric coupling are not powerful and solid.

(1) The reviewer can't agree with the response for comment 4. It should be clarified why the positive and negative pyrocurrents coexist in the same data line of Fig. 4b.

(2) To verify the magnetoelectric coupling in Mn_2MnWO_6 , the temperature dependence of pyrocurrent and pyroelectric polarization under different magnetic field is essential. As shown in Fig. 4b, is the difference between the pyrocurrent under 0 and 1 T significantly larger than the error bars? The reviewer strongly suggests that the temperature dependence of pyrocurrent under higher magnetic fields should be provided. Moreover, the reviewer does not agree with the response of comment 5. We should remind the authors that from Fig. 2c they give, the magnetization is not saturated even at 15 T.

(3) The measurement of magnetic field dependent of polarization for Mn_2MnWO_6 at 20 K in Supplementary Fig. S16 does not give clear and convincing data to clarify the existing of magnetoelectric coupling in Mn_2MnWO_6 . Perhaps there are background noises due to the thermal fluctuation.

(4) In order to exclude the possibility of charge accumulation in the measurement, which always leads to the false impression in the magnetoelectric coupling measurement with pyro-current method, more measurements are necessary for recording pyrocurrent in different warming rate as a function of temperature between 10 and 80 K, such as 1K/min, 3K/min and 5K/min.

Reviewer #2 (Remarks to the Author):

I am satisfied by the answers from the authors to my comments. The added discussion clearly highlights the specific properties of Mn_2MnWO_6 and the general interest of such "NTO materials".

Reviewer #1 (Remarks to the Author):

Even though the revised manuscript has corrected most of the comments, the evidences for magnetic induced polarization and magnetoelectric coupling are not powerful and solid.

(1) The reviewer can't agree with the response for comment 4. It should be clarified why the positive and negative pyrocurrents coexist in the same data line of Fig. 4b.

Authors' response

The pyrocurrent measured on a constant heating rate is directly proportional to the derivative of the $P(T)$ curve. The $P(T)$ gained from the integration of the pyrocurrent and from refinement of the lattice parameters both yield a qualitatively similar curvature: starting from low T the polarization first increases (-> positive derivative, positive pyrocurrent), then experiences a sharp drop (-> sharp negative peak in the pyrocurrent), then rises again (-> positive pyrocurrent). While this behavior is unusual in ferroic systems with monotonous temperature dependence of the order parameter. The qualitative agreement between both experimental methods shows that our system is obviously more subtle and further theoretical work on this might be interesting in the future.

(2) To verify the magnetoelectric coupling in Mn_2MnWO_6 , the temperature dependence of pyrocurrent and pyroelectric polarization under different magnetic field is essential. As shown in Fig. 4b, is the difference between the pyrocurrent under 0 and 1 T significantly larger than the error bars? The reviewer strongly suggests that the temperature dependence of pyrocurrent under higher magnetic fields should be provided. Moreover, the reviewer does not agree with the response of comment 5. We should remind the authors that from Fig. 2c they give, the magnetization is not saturated even at 15 T.

Authors' response

In principle the referee is right that in order to study the details of a magneto-electric coupling more measurements would be needed. However, as discussed in the paper, for high fields the magnetic order is not altered anymore and the $M(H)$ data only show a linear magnetic field dependence that is probably related to a continuous spin-canting within an antiferromagnetic spin structure. Therefore, we would not expect further qualitative changes in the induced polarization. **A corresponding result is also derived by the new permittivity measurements in magnetic field: In higher fields the feature in $\epsilon_{ps}(T)$ decays (Fig. 4d).** Also, we have to emphasize that subtle, magnetic field induced changes in the pyro-current signatures will be hampered by the uncertainty in the temperature determination. Due to the high

heating rates needed for the pyro-current measurements such T -uncertainty is unavoidable and would weaken any conclusions concerning the details of the (M,T) phase boundary. Furthermore, we want to remind that the absolute values for the polarization analysis are severely hampered by the polycrystalline nature of our samples.

(3) The measurement of magnetic field dependent of polarization for Mn_2MnWO_6 at 20 K in Supplementary Fig. S16 does not give clear and convincing data to clarify the existing of magnetoelectric coupling in Mn_2MnWO_6 . Perhaps there are background noises due to the thermal fluctuation.

Authors' response

We have done further temperature dependent dielectric measurements at several magnetic fields from 0 to 10 T and observed anomalies around the magnetic transition temperature as present in the updated **Fig. 4d** shown below. The shift of the transition temperature with the magnetic field as well as the observed suppression in high magnetic fields clearly convince magnetoelectric coupling in Mn_2MnWO_6 .

“**Fig. 4 (d)** Temperature dependent dielectric data between 0 and 10 T show anomalies around T_N and indicate magnetoelectric coupling”

(4) In order to exclude the possibility of charge accumulation in the measurement,

which always leads to the false impression in the magnetoelectric coupling measurement with pyro-current method, more measurements are necessary for recording pyrocurrent in different warming rate as a function of temperature between 10 and 80 K, such as 1 K/min, 3 K/min and 5 K/min.

Authors' response

We have carried out measurements with warming rates of 1, 2, and 3 K/min, respectively, and the results are shown in the figure below (and added to the Supporting Information as **Figure S16** in Page S25), which qualitatively confirm the expected scaling of the current with the rate of temperature change and thus exclude the possibility of ohmic, i.e. current related phenomena. (However the T -shift of the signature has to be attributed to the insufficient thermal coupling and thus denotes the experimental problems to determine e.g. a subtle shift of the feature in magnetic field as discussed above.) Also, the additional dielectric measurements now shown in Figure 4d demonstrate that we indeed observe a change in polarization as this ac measurements are not subject to charge accumulation.

“Figure S16. Temperature dependent pyrocurrent measurements on Mn_2MnWO_6 in different warming rate of 1, 3, and 5 K/min, respectively, which qualitatively confirm the expected scaling of the current with the rate of temperature change and thus exclude the possibility of ohmic, i.e. current related phenomena. (However the T -shift of the signature has to be attributed to the insufficient thermal coupling and thus denotes the experimental problems to determine e.g. a subtle shift of the feature in magnetic field as discussed above.) Also, the additional dielectric measurements now shown in Figure 4d demonstrate that we indeed observe a change in polarization as this ac measurements are not subject to charge accumulation.”

Reviewer #2 (Remarks to the Author):

I am satisfied by the answers from the authors to my comments. The added discussion clearly highlights the specific properties of Mn_2MnWO_6 and the general interest of such "NTO materials".

Reviewers' Comments:

Reviewer #1 (Remarks to the Author):

The authors have replied to the reviewer's comments. Unfortunately, after reading the response letter, I find that the evidences of multiferroic and magnetoelectric properties in this sample are not powerful. More substantial experimental data and further studies are needed before this manuscript can be considered for publication. My further discontent can be concluded as follows:

- (1) The authors don't give a convictive explanation for the origin of the positive and negative pyrocurrents.
- (2) As shown in Fig.S16, the peak of pyrocurrent shifts remarkably with different warming rates. This phenomenon further demonstrates that this sample may not have intrinsic ferroelectric polarization and the pyrocurrent comes from the trapped charges.
- (3) The authors don't give the experimental results of the temperature dependence of pyrocurrent and pyroelectric polarization under different magnetic field as the reviewer requested. They should know that magnetodielectric effect is not equivalent to magnetoelectric effect.

Reviewer #4 (Remarks to the Author):

The authors reported a compound Mn_2MnWO_6 prepared under high pressure and high temperature conditions. This material is well characterized by many pertinent techniques and shows appealing properties which might be of interest in the field of multiferroics. The crystal and spin structures, polar nature, and magnetism of the title compound are unambiguously presented by comprehensive measurements. Regarding to the magnetoelectric and pyroelectric behaviors, the authors do not show thoroughly studies by only presenting qualitative images of the weak pyro-response and small switchable ferroelectric components. It is understandable that no one can expect a strong ME effect from polycrystalline samples. Therefore, more work on single crystal or epitaxial thin film is essential to quantitatively determine ME coupling in the future.

Concerning the comments from Reviewer #1:“(1) The authors don't give a convictive explanation for the origin of the positive and negative pyrocurrents.”It is hard to precisely correlate the pyrocurrent results with in-situ temperature changing according to the measurements in this work. But to my experience, it is understandable that the pyrocurrent signals root in the temperature-dependent-variation of polarization. As can be seen in Fig. 4a, the slope of the P_s versus T curve shows negative/positive crossover(s) between 10-80 K, thus giving positive and negative pyrocurrent values.“(2) As shown in Fig. S16, the peak of

pyrocurrent shifts remarkably with different warming rates. This phenomenon further demonstrates that this sample may not have intrinsic ferroelectric polarization and the pyrocurrent may come from the trapped charges.”I agree with Reviewer #1 at this point if only taken Figure S16 into account. However, Figures 4c-d, 5, and S17 can provide the evidence of intrinsic ferroelectric polarization, although not robust in such a polycrystalline sample.”“(3) The authors don’t give the experimental results of the temperature dependence of pyrocurrent and pyroelectric polarization under different magnetic field as the reviewer requested. They should know that magnetodielectric effect is not equivalent to magnetoelectric effect.”The authors show the results at 0 and 1T in Figure 4c, more measurements under different magnetic field may show a clearer image here. However, in Fig. 2c a linear change of magnetization was observed as H field is above 2 T. Therefore, I doubt we won’t be able to get any useful information from further measurements. As mentioned in (2), the measured results are qualitative, that is sufficient to make this story. This work seems to be an extension of authors’ previous work in A2BB’O6 family. Though the findings in Mn₂MnWO₆ are little bit far from perfect, I believe this work still push forward a lot in single phase multiferroics. In compared with previous research on this topic, based on my knowledges, this is the second report on quantitative determine ME coupling in this type materials. (The first one was reported by Sang-Wook Cheong et al in Nature Communications in 2014 (Ref. 5) Overall, this is an interesting paper, and worthy of publication in Nature Communications.

Reviewer #5 (Remarks to the Author):

The manuscript "Magnetostriction-Polarization Coupling in Multiferroic Mn₂MnWO₆" by Li and coworkers describes in detail the different ferroic properties of the corundum derivative Mn₂MnWO₆ and how they couple. The authors present a detailed structural study which is correlated to the magnetism and magnetic structure of the material, followed by dielectric (pyrocurrent, C(T), PFM) measurements, and they discuss briefly second harmonic measurements. Overall the study is very extensive and well done, and in particular the magnetic measurements well presented and discussed. The manuscript version I have read is the already revised version which includes the temperature dependent dielectric measurements taken at different applied magnetic fields. From my point of view, the authors have demonstrated that Mn₂MnWO₆ is ferroelectric (PFM) and AFM (M(T)) at the same time below approx. 60K and the AFM transition is correlated to a change in the Mn₂-Mn₃ bondlength. The C(T,H) measurements further show that a magnetic fields clearly influences the maximum of the dielectric response (Fig. 4d) which is indicative of a coupling of the magnetic field to the electric dipole moment. Therefore, Mn₂MnWO₆ can be called a multiferroic material with magneto-electric coupling. In my opinion, the manuscript should be published in NatComms but I have a

couple of comments the authors should consider prior publication.

My points are the following.

The authors show SHG data up to 800K with a finite intensity (Fig. S15). As correctly pointed out, the second harmonic signal is indicative of a symmetry breaking of the crystalline symmetry. But the SHG signal is only associated with a polar resonance which does not mean it is FE. Mn_2MnWO_6 is therefore in a polar state already well above RT. What SHG cannot show is FE. For this, the authors would have to apply an electric field with different polarities and measure the SHG response. In any case, the SHG signal also represents an order parameter which increases very slowly with decreasing temperature before decreasing at around the temperature where the Mn²-Mn³ bond length is changing and increases again below approx 20K. This is a rather odd behaviour for an order parameter and could indicate different polar ordering. I also do not really understand the sentence: The magnetostriction-polarization coupling around TN is also visible in the fluctuation of the SHG intensity (Supplementary Fig. S15). How do you deduce the coupling from these measurements since you do not show the influence of a magnetic or electric field on the polar response of the materials system?? You just show the plain SHG vs T signal.

This brings me to the PFM measurements, I couldn't find in the manuscript at which temperature the PFM measurements were taken. Maybe I overlooked it. You do show, however, P(V) loops taken at different temperatures (Fig. S20). Apart from the 200K measurement which looks a little like the banana P(V) measurement of James Scott's paper, the other measurements look more convincing and would go along with the polar response of the SHG measurements. Meaning Mn_2MnWO_6 would be already FE well above TN. That you have a poling, you do show with your PFM measurements. From this point of view, you demonstrate with these measurements FE and the pyrocurrent measurement are not really needed. They do show, however, that something in the polar response of the materials system below TN is changing whatever the strange change in the current response means. What I would not understand in this context, why the pyrocurrent in Fig. 4b is almost zero above 50K. This would be some kind of contradiction. I know, the currents you measure are very small and this is non-trivial to do and the polycrystalline nature of the sample is certainly not helping a lot with these measurements. Also from this point of view, the pyrocurrent measurements are the least convincing ones with respect to the more clear-cut PFM images or the P(V) loops.

You can leave the manuscript as it is, but some restructuring without changing the content may be helpful and more convincing to a reader.

Point-by-point response to Reviewers' comments for manuscript:

NCOMMS-17-02861B

Title: Magnetostriction-Polarization Coupling in Multiferroic

Authors: ML Li et al

Mn₂MnWO₆"**REVIEWERS' COMMENTS:**

Reviewer #1 (Remarks to the Author):

The authors have replied to the reviewer's comments. Unfortunately, after reading the response letter, I find that the evidences of multiferroic and magnetoelectric properties in this sample are not powerful. More substantial experimental data and further studies are needed before this manuscript can be considered for publication. My further discontent can be concluded as follows:

- (1) The authors don't give a convictive explanation for the origin of the positive and negative pyrocurrents.
- (2) As shown in Fig.S16, the peak of pyrocurrent shifts remarkably with different warming rates. This phenomenon further demonstrates that this sample may not have intrinsic ferroelectric polarization and the pyrocurrent comes from the trapped charges.
- (3) The authors don't give the experimental results of the temperature dependence of pyrocurrent and pyroelectric polarization under different magnetic field as the reviewer requested. They should know that magnetodielectric effect is not equivalent to magnetoelectric effect.

Authors' response

We thank Reviewer #1 for taking the time to read our responses to the previous comments on our paper. We respectfully disagree with the continued negative analysis of our results, which are contradictory to the comments of Reviewers #4 and 5.

We thank Reviewers #4 and 5 for their thoughtful comments and for supporting our work/paper for publication in Nature Commun. Below we respond to their comments

point by point:

Reviewer #4 (Remarks to the Author):

The authors reported a compound Mn_2MnWO_6 prepared under high pressure and high temperature conditions. This material is well characterized by many pertinent techniques and shows appealing properties which might be of interest in the field of multiferroics. The crystal and spin structures, polar nature, and magnetism of the title compound are unambiguously presented by comprehensive measurements. Regarding to the magnetoelectric and pyroelectric behaviors, the authors do not show thoroughly studies by only presenting qualitative images of the weak pyro-response and small switchable ferroelectric components. It is understandable that no one can expect a strong ME effect from polycrystalline samples. Therefore, more work on single crystal or epitaxial thin film is essential to quantitatively determine ME coupling in the future.

Authors' response

We agree with this careful analysis of our paper, and that more work on single crystal or epitaxial thin film is essential on Mn_2MnWO_6 . It is well known that the growth of single crystal or epitaxial film of high-pressure-made phase is a challenge. This is planned in the future.

Concerning the comments from Reviewer #1:

“(1) The authors don't give a convictive explanation for the origin of the positive and negative pyrocurrents.” It is hard to precisely correlate the pyrocurrent results with in-situ temperature changing according to the measurements in this work. But to my experience, it is understandable that the pyrocurrent signals root in the temperature-dependent-variation of polarization. As can be seen in Fig. 4a, the slope of the P_s versus T curve shows negative/positive crossover(s) between 10-80 K, thus

giving positive and negative pyrocurrent values.

Authors' response

We agree with Reviewer #4 at this point.

“(2) As shown in Fig. S16, the peak of pyrocurrent shifts remarkably with different warming rates. This phenomenon further demonstrates that this sample may not have intrinsic ferroelectric polarization and the pyrocurrent may come from the trapped charges.” I agree with Reviewer #1 at this point if only taken Figure S16 into account. However, Figures 4c-d, 5, and S17 can provide the evidence of intrinsic ferroelectric polarization, although not robust in such a polycrystalline sample.

Authors' response

We agree with Reviewer #4.

“(3) The authors don't give the experimental results of the temperature dependence of pyrocurrent and pyroelectric polarization under different magnetic field as the reviewer requested. They should know that magnetodielectric effect is not equivalent to magnetoelectric effect.” The authors show the results at 0 and 1T in Figure 4c, more measurements under different magnetic field may show a clearer image here. However, in Fig. 2c a linear change of magnetization was observed as H field is above 2 T. Therefore, I doubt we won't be able to get any useful information from further measurements. As mentioned in (2), the measured results are qualitative, that is sufficient to make this story.

Authors' response

We agree with Reviewer #4. Thus more measurements at higher magnetic field (raised by Reviewer #1) are not necessary.

This work seems to be an extension of authors previous work in $A_2BB'O_6$ family.

Though the findings in Mn_2MnWO_6 are little bit far from perfect, I believe this work still push forward a lot in single phase multiferroics. In compared with previous research on this topic, based on my knowledges, this is the second report on quantitative determine ME coupling in this type materials. The first one was reported by Sang-Wook Cheong et al in Nature Communications in 2014 (Ref. 5)

Overall, this is an interesting paper, and worthy of publication in Nature Communications.

Reviewer #5 (Remarks to the Author):

The manuscript "Magnetostriction-Polarization Coupling in Multiferroic Mn_2MnWO_6 " by Li and coworkers describes in detail the different ferroic properties of the corundum derivative Mn_2MnWO_6 and how they couple. The authors present a detailed structural study which is correlated to the magnetism and magnetic structure of the material, followed by dielectric (pyrocurrent, $C(T)$, PFM) measurements, and they discuss briefly second harmonic measurements. Overall the study is very extensive and well done, and in particular the magnetic measurements are well presented and discussed. The manuscript version I have read is the already revised version which includes the temperature dependent dielectric measurements taken at different applied magnetic fields. From my point of view, the authors have demonstrated that Mn_2MnWO_6 is ferroelectric (PFM) and AFM ($M(T)$) at the same time below approx. 60 K and the AFM transition is correlated to a change in the Mn2-Mn3 bond length. The $C(T,H)$ measurements further show that a magnetic fields clearly influences the maximum of the dielectric response (Fig. 4d) which is indicative of a coupling of the magnetic field to the electric dipole moment. Therefore, Mn_2MnWO_6 can be called a multiferroic material with magneto-electric coupling. In my opinion, the manuscript should be published in Nat. Commun. but I have a couple of comments the authors should consider prior publication.

My points are the following.

The authors show SHG data up to 800 K with a finite intensity (Fig. S15). As correctly pointed out, the second harmonic signal is indicative of a symmetry breaking of the crystalline symmetry. But the SHG signal is only associated with a polar response which does not mean it is FE. Mn_2MnWO_6 is therefore in a polar state already well above RT. What SHG cannot show is FE. For this, the authors would have to apply an electric field with different polarities and measure the SHG response. In any case, the SHG signal also represents an order parameter which increases very slowly with decreasing temperature before decreasing at around the temperature where the Mn2-Mn3 bond length is changing and increases again below approx 20 K. This is a rather odd behaviour for an order parameter and could indicate different polar ordering. I also do not really understand the sentence: The magnetostriction-polarization coupling around T_N is also visible in the fluctuation of the SHG intensity (Supplementary Fig. S15). How do you deduce the coupling from these measurements since you do not show the influence of a magnetic or electric field on the polar response of the materials system?? You just show the plain SHG vs T signal.

Authors' response

The referee is correct, the second harmonic signal is indicative of a noncentrosymmetric material, though it is not proof of ferroelectricity. The finite SHG response shows that the material is noncentrosymmetric over the entire measured temperature range. As the referee correctly indicates, the gradually increasing SHG signal with decreasing temperature is due to the order parameter related to long-range polar order, a quantity that is expected to increase with decreasing temperature in polycrystalline ferroelectrics.

The behavior of the SHG response at around 60 K indicates the occurrence of a phase transition, which is consistent with the onset of AFM order at the Neel temperature of 58 K. While this response cannot be used in isolation as evidence of

magnetostriction-polarization coupling, since SHG depends non-trivially on both the crystal symmetry (which is affected by the magnetic ordering) as well as degree of noncentrosymmetry (which is affected by the polarization), and so such a coupling is expected to be reflected as an anomaly in the SHG signal, something that is indeed observed. Further, such a behavior has previously been shown to be associated with spin-charge coupling (Ramirez, M. O., *et al. Phys. Rev. B* 79, 1–9, 2009, Ref. 25 in the updated Supplementary References) in thin films.

Therefore, the following sentences have been added at the end of Supplementary Note 3 in Page 30 of Supplementary Information:

“The fluctuation of the SHG response in Supplementary Fig. 15 at around T_N indicates the presence of spin-charge coupling, as shown in previous works.[25] This behavior is consistent with the magnetostriction-polarization coupling that is proposed.”

This brings me to the PFM measurements, I couldn't find in the manuscript at which temperature the PFM measurements were taken. Maybe I overlooked it. You do show, however, P(V) loops taken at different temperatures (Fig. S20). Apart from the 200 K measurement which looks a little like the banana P(V) measurement of James Scott's paper, the other measurements look more convincing and would go along with the polar response of the SHG measurements. Meaning Mn_2MnWO_6 would be already FE well above T_N . That you have a poling, you do show with your PFM measurements. From this point of view, you demonstrate with these measurement FE and the pyrocurrent measurement are not really needed. They do show, however, that something in the polar response of the materials system below T_N is changing whatever the strange change in the current response means.

Authors' response

PFM was all performed at room temperature as is stated on pg. 9, line 11 (and was overlooked by this Reviewer). This is highlighted again in the Methods section.

We fully agree with the referee that the $P(E)$ -loops above 150 K are dominated by conductivity – maybe not by ionic conductivity as it is the case for Scott’s bananas, but by thermally activated electronic conductivity. Therefore, the 200 K-curve is only shown in Supplementary Fig. 20 to demonstrate that no ferroelectric switching behavior can be evaluated from $P(E)$ -measurements for high temperatures, due to this masking by conductivity. This is indeed different for the PFM measurements and therefore they are more reliable in this case.

What I would not understand in this context, why the pyrocurrent in Fig. 4b is almost zero above 50 K. This would be some kind of contradiction. I know, the currents you measure are very small and this is non-trivial to do and the polycrystalline nature of the sample is certainly not helping a lot with these measurements. Also from this point of view, the pyrocurrent measurements are the least convincing ones with respect to the more clear-cut PFM images or the $P(V)$ loops.

Authors’ response

Indeed, the pyro-current measurements can only give a qualitative picture due to the polycrystalline nature of the sample. Nevertheless, the peak-like response near T_N corresponds to the change in the polarization also recovered from the structural refinement (Fig.4a). Above this temperature only the slight and continuous changes in the polar structure due to thermal expansion survive; therefore, the pyro-current $I = dP/dT$ has to be small above 60 K where the temperature dependence observed in the structural refinement data is flat. The referee is right, that from the pyrocurrent measurements alone it would not be satisfactorily convincing to prove the magnetic signature in the temperature dependence of the polarization. However, the qualitative correspondence between the $P(T)$ results from both pyro-current and structural refinement paints a convincing overall picture.

You can leave the manuscript as it is, but some restructuring without changing the content may be helpful and more convincing to a reader.